# Predicting the evolution of *Escherichia coli* by a data-driven approach

Xiaokang Wang[1,2], Violeta Zorraquino[2], Minseung Kim[2,3], Athanasios Tsoukalas[2] & Ilias Tagkopoulos [2,3]

A tantalizing question in evolutionary biology is whether evolution can be predicted from past experiences. To address this question, we created a coherent compendium of more than 15,000 mutation events for the bacterium *Escherichia coli* under 178 distinct environmental settings. Compendium analysis provides a comprehensive view of the explored environments, mutation hotspots and mutation co-occurrence. While the mutations shared across all replicates decrease with the number of replicates, our results argue that the pairwise over-lapping ratio remains the same, regardless of the number of replicates. An ensemble of predictors trained on the mutation compendium and tested in forward validation over 35 evolution replicates achieves a 49.2 ± 5.8% (mean ± std) precision and 34.5 ± 5.7% recall in predicting mutation targets. This work demonstrates how integrated datasets can be harnessed to create predictive models of evolution at a gene level and elucidate the effect of evolutionary processes in well-defined environments.

[1] Department of Biomedical Engineering, University of California, Davis, Davis, CA 95616, USA. [2] Genome Center, University of California, Davis, Davis, CA 95616, USA. [3] Department of Computer Science, University of California, Davis, Davis, CA 95616, USA. These authors contributed equally: Xiaokang Wang, Violeta Zorraquino. Correspondence and requests for materials should be addressed to I.T. (email: itagkopoulos@ucdavis.edu)

t is well known that the result of evolutionary processes in a population is not random for a given environment. However, our understanding of how reproducible the fixated mutation patterns are and at what degree they can be predicted is limited[1–3]. Given the random and complex nature of evolutionary adaptation that encompasses both local and global effects, it is unclear whether individual mutation targets can be associated with confidence to a particular environment. One such signal is the mutation of a common group of genes that govern global regulation under a variety of environmental conditions[4–6]. While evolution is a mixed result of deterministic and stochastic events, there have been several studies that aim to predict the evolution of a trait. Early on, in vitro evolution experiments were used to predict the emergence of antibiotic resistance[7], while engineered *Escherichia coli* cells have demonstrated predicted fitness after multiple rounds of evolution[8].

Adaptive laboratory evolution (ALE) has been applied to elucidate the genetic basis and potential of evolutionary adaptation[9,10]. One of the preferred hosts for ALE experiments is *E. coli*, due to its well-studied physiology and cellular organization, short generation time (typically 8–10 generations per day), small genome (5 M bases) and high relevance to biotechnology and health. Most of ALE experiments are performed under stress conditions, such as ethanol[11], antibiotics[12–17] and high acidity[9], while the number of generations vary widely from a few dozens to thousands. Inexpensive resequencing of the evolved clones by next-generation sequencing has provided insights into the genetic basis of acquired fitness in novel environments[18–21] as well as the variability of evolved cell populations during evolution[22–24].

When it comes to prediction, machine learning has been successfully applied in a variety of topics, from classification of the binding activity of proteins[25] and to DNA[26] to guiding protein design[27]. Data from genome-wide association studies have been used in various studies including for calculation of the gene mutation probability and the role of pleiotropy in adaptation. To apply similar methods in this context, one has to integrate the evolutionary data in a way that they can be mined, reveal patterns and test hypotheses, similar to what is done for omics data[28–30]. Mutation databases currently exist[31–33], but they are not suitable for training machine learning models due to lack of environmental metadata, limited focus and size. If we aspire to predict evolution at some degree, efforts need to be more systematic, larger in size and focused on creating the necessary infrastructure of well-integrated data and methods. Being able to make accurate predictions about the number, type and position of mutations will lead to an improved understanding of how organisms evolve and will allow us to design better experiments. Conversely, we can use the same datasets to build predictors of the environment in which an organism lives or has evolved, given its mutation profile[34]. Predicting genetic mutation targets accurately and reproducibly will be a boon to research and industrial applications, as a critical challenge is traversing the vast combinatorial space in the case of strain engineering or fermentation settings for microbial biotechnology. Most methods so far focus on descriptive analysis versus techniques that produce predictive or prescriptive insights[3]. In all cases, each additional dataset that will be integrated to the current compendium will contribute towards the accurate prediction of mutation targets in novel environments.

In this study, we curated the current scientific literature for publications with *E. coli* whole-genome sequencing (WGS) data and necessary metadata (Fig. 1a). We then analyzed this mutation compendium of more than 15,000 events and we organized their corresponding conditions and attributes so that it can be used as a training set. Then we used it to train "evolution" predictors that have the capacity to predict gene mutation targets, at gene (not nucleotide) level, given a novel environmental setting (Fig. 1b).

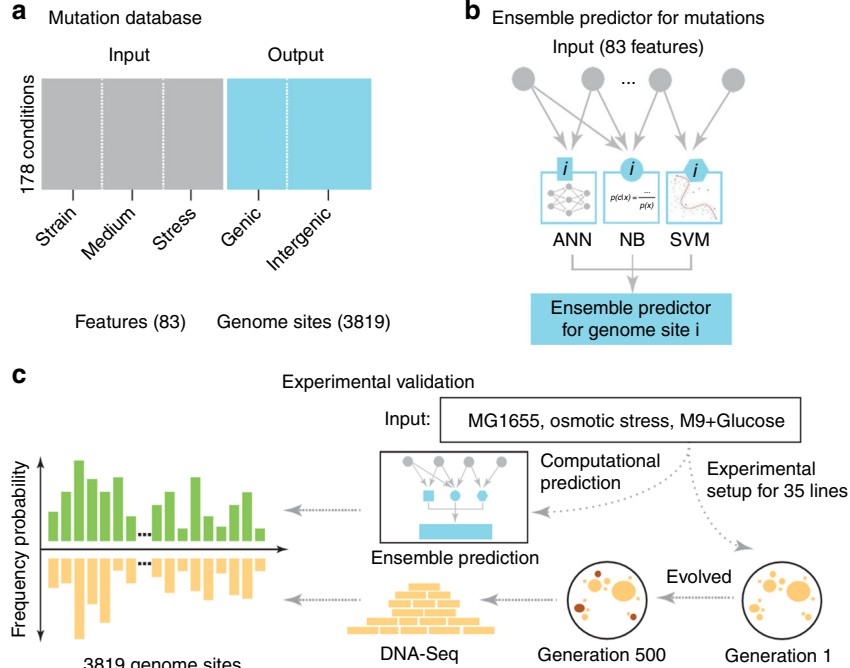

**Fig. 1** Overview of predicting mutations in *E. coli* using a data-driven approach. **a** A compendium was constructed with mutation profiles across 178 conditions over 83 features that capture attributes related to the strain, medium and stress from experiments reported in 95 publications. **b** We built three individual predictors, namely an Artificial Neural Network (ANN), Support Vector Machines (SVM) and a Naive Bayes (NB) model, which are integrated under one Ensemble method. **c** Assessment of the predictions from all three individual predictors and the Ensemble method is performed through forward validation over a novel experimental setting through the evolution and whole-genome resequencing of 35 cell lines

As a forward validation step, we evolved 35 independent *E. coli* lines for 500 generations, identified the mutations and then compared the computational predictions with our experimental findings (Fig. 1c). Compendium analysis provides a comprehensive view of the explored environments, mutation hotspots and mutation co-occurrence. Interestingly, our data show that fixation rate follows a linear relationship that exhibits non-linearity as the number of generations grow due to the emergence of hypermutator cells. We find that a linear relationship exists between the averaged frequency of all the mutations under a specific condition and the number of biological replicates. While the mutations shared across all replicates decrease with the number of replicates, as expected, our results argue that the same ratio across any two replicates remain the same, regardless of the number of replicates, which is quite unexpected. We observe a higher likelihood to hit DNA-related functions in hypermutators, which is analogous to what has been observed recently in cancer cells with high mutation rates. An ensemble of predictors trained on the mutation compendium and tested in forward validation over 35 evolution replicates achieves a $49.2 \pm 5.8\%$ precision and $34.5 \pm 5.7\%$ recall in predicting mutation targets in the novel condition.

## Results

**The mutation database of *E. coli*.** A total of 95 papers containing ALE experiments were analyzed to extract the adaptive mutations published in evolutionary *E. coli* studies (Supplementary Data 1). Out of these 95 papers, 41 had available sequencing data that were compiled in a database of 15,402 mutation events grouped in 3819 genomic sites across 574 evolved genomes (Supplementary Data 2). The data have been collected across 178 culture conditions with the number of replicates ranging from 1 to 115. Each evolution condition was defined by three attributes: strain, medium and stress (Fig. 2a). For each mutation event, we recorded its genome position with respect to a reference genome and the mutation event type. Among the 15,402 mutations, 8759 were single-nucleotide polymorphisms (SNPs; 57%), 3994 were deletions (26%), 1342 were insertions (9%), 1303 were amplifications (8%) and 4 were inversions (0.02%) (Fig. 2b). In terms of spatial location, the 15,403 mutation events hit 3819 mutation sites, with 3065 (80.3%) and 754 (19.7%) of them in coding and intergenic regions, respectively.

**Mutation hotspots and key targets across the *E. coli* genetic landscape.** Next, we mapped the frequency and position of mutation hotspots in the *E. coli* genome. The mutation frequency for each locus in the 178 unique conditions follows a power law distribution with a long tail, having the top 0.5% genome sites accounting for the 5% of all mutations (Fig. 2c, Supplementary Data 3). Gene ontology (GO) enrichment showed that 12 out of the top 20 genes most likely to be hit by a mutation are involved in carbohydrate transport and metabolism, an adaptation to the carbon source in the media (Supplementary Fig. 1). The most frequently mutated site found is the RNA polymerase subunit *rpoB* which is detected in 30.9% of the conditions present in this study. *rpoB* hits are ubiquitous in evolution experiments, such as the evolution of antibiotic resistance. Interestingly, recent studies report that antibiotic-resistant clones based on *rpoB* mutations emerge early in the evolution, even in the absence of antibiotics in the environment[35], and also confer increased evolvability[36]. Other sigma factors like *rpoS* and *rpoC* are also mutation targets[4,37], as a single mutation in the transcription machinery can lead to substantially different metabolic products[38]. The other two of the top three genes are *pyrE-rph* (49/178) and *pykF* (37/178) that have been associated with adaptation to media and adverse conditions, such as the presence of antibiotics and oxidative stress[39].

We used a 5 kb sliding window along the *E. coli* MG1655 genome to find regions that were most or least likely to be hit by a mutation. The distribution of mutation counts has a long tail that we fit with a gamma distribution. Hotspots were identified as the top 5% regions with respect to mutations (Fig. 2d). Similarly, the genomic areas that were never hit by a mutation were flagged as depletion spots (Fig. 2e, Supplementary Data 3). Functional analysis of the top five hotspots argues that metabolic genes and genes related to transport are more prone to mutations: the top five GO terms are carbohydrate catabolic process, magnesium ion binding and nucleotidyltransferases (Supplementary Fig. 2). In contrast, depletion spots mostly contain genes associated with critical structural and biosynthesis processes, such as organelle proteins, membrane, cell wall and amino acid synthesis (Supplementary Fig. 3).

In our previous analysis, we did not take into account the 36 hypermutator strains, strains with a mutation rate larger than 0.1 mutations per genome duplication that have been identified as such. Hypermutators are generally expected to follow a different trajectory during evolution, but it is unclear if their mutation signature overlaps with that of other mutated strains. We found that the mutation frequencies of the most frequently hit genes in hypermutator strains are higher than in other strains (52.9% vs 29.8% in Table 1). Nucleotide binding is still the most prominent function of the target genes in hypermutator strains, but at lower frequency (6 vs 10 out of 20), while *rpoB* continues to be a popular mutation target. Interestingly, the DNA gyrase subunit B that negatively supercoils DNA to maintain it in an underwound state is the most common genomic site harboring a mutation[40,41]. We observe a higher likelihood to hit DNA-related functions in hypermutators, which is analogous to what has been observed recently in cancer cells with high mutation rates[42] (among the 36 hypermutators, 100% have at least one DNA-related gene mutated. In contrast, the percentage is 51% for normal evolved replicates; Supplementary Data 4).

**Co-occurring mutations.** To investigate any significant overlap or dependencies among mutations, we performed an analysis of the pairwise association in mutation events using a spectral clustering approach. Interestingly, we consistently detected six clusters, each with a distinct functional signature (Fig. 3a, Supplementary Data 5). We analyzed the enriched molecular function GO terms in each cluster (Fig. 3c, a *p* value of less than 0.1 was used as a cut-off for GO enrichment). In the case of GO cellular components (Supplementary Fig. 4), clusters 2, 3, 4 and 6 posses a high number of membrane proteins (11/24, 33/76, 20/63 and 31/163 respectively). Moreover, 6 of the 33 membrane proteins in cluster 3 are involved in the transport. The only pathway enriched in any of the clusters is the two-component system, which is present in 9 genes in cluster 1 and 8 in cluster 4. Two-component systems sense the environmental conditions and activate specific pathways in the bacteria, which are clear targets for adaptation under selection pressure. The top co-occurring mutation pairs are not inside any cluster and have two genes involved, *gntU* and *yhjN*, a low-affinity gluconate transporter and biofilm formation precursor respectively (Supplementary Data 6). Mutations in these genes are a general response to improve carbon acquisition and leads to a reduced ability of biofilm formation[43]. Other genes present in these pairs are involved in transcription and transport of nutrients. We also performed a network enrichment analysis by building a network of mutation associations for the five most popular stresses in our database: heat, anaerobic growth, acid, presence of antibiotics and butanol (Supplementary Fig. 5). The analysis depicts patterns that are both environment specific (e.g., we found that mutations linked to anaerobic growth are prone to be found in other stresses too) and

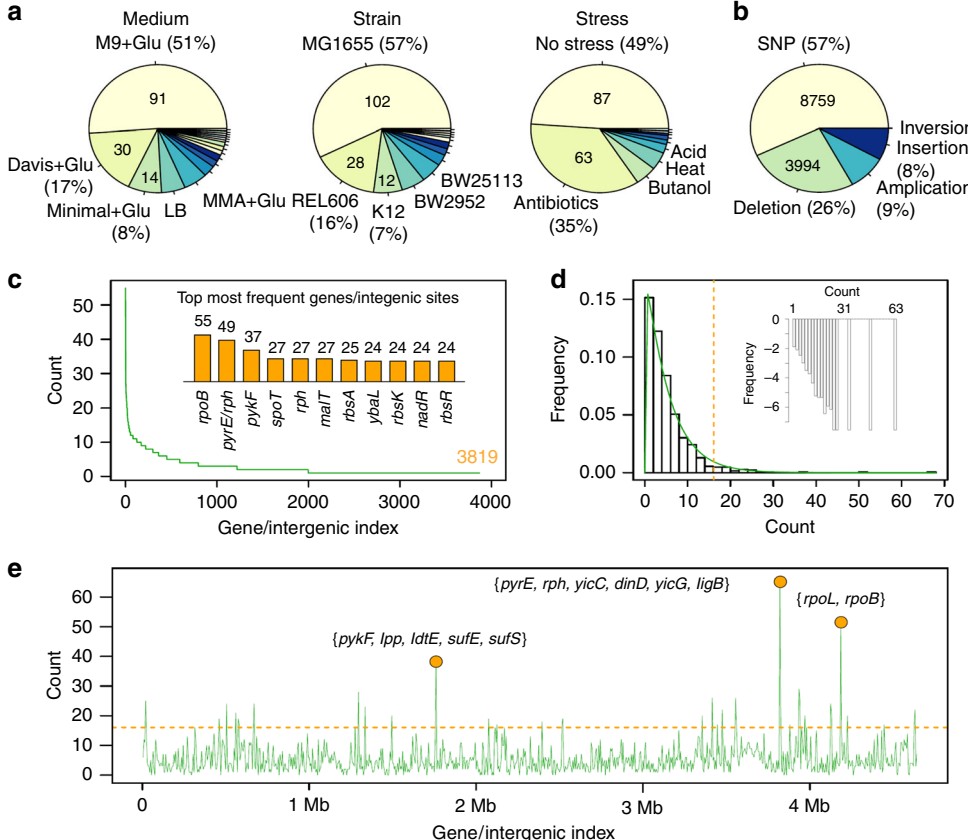

**Fig. 2** Database summary and statistics. **a** The distribution of culture conditions in terms of medium, strain and stress. In the case of medium, the legend is restricted to evolution runs with multiple replicates to save space. **b** The frequencies of different mutation types in the database. **c** The distribution of mutations among 3819 genome sites with the top genes presented inside the plot. **d** The frequency of the genomes sites hit by a mutation under 178 culture conditions. The count distribution for each genome site (log y-axis) is shown in the inset. The histogram is approximated by a gamma distribution (green curve). Dotted orange line depicts $p$ value < 0.05. **e** All the mutations were presented along the genome of MG1655 *E. coli.* with the top three hotspots flagged. When visualizing the mutations along the genome, we used a 5 kb sliding window to find regions that were most/least likely to be hit by a mutation. Dotted orange line depicts $p$ value < 0.05

gene specific (e.g., *rph* mutations exist in three stress environments and it is not related to media adaptation).

**Fixation rate and number of generations.** To understand the relationship of the fixed mutation rates and number of generations, we plotted the number of fixed mutations found in each evolved clone against the generations elapsed with and without hypermutator clones (Fig. 3b). Our analysis shows that mutation rate is linear without hypermutations, and exponential if hypermutations are included. To account for other possible factors, such as strain, medium and stress, we used analysis of variance (ANOVA) (Supplementary Information and Supplementary Fig. 6a). We found that with hypermutators this significant difference in fixation rates between long and intermediate/short-term evolution runs (cut-off at 30,000 generations) exists, even when variation (or lack of) for strain, medium or stress was taken into account ($p$ value = $1.28 \times 10^{-3}$). For the four categories of mutations, synonymous SNP, non-synonymous SNP, insertion and deletion among the non-hypermutator clones, the fixation rate is also linear (Supplementary Fig. 7). For hypermutators, mutation rate ranges from 0.035 to 1.5 with a mean of 0.26 per generation. The number of mutations increased linearly with the generation elapsed. For normal clones, the mutation rate varies from 0.0018 to 0.3 with a median of 0.0068 per generation per genome, in accordance and in confirmation of Drake's landmark observation to a mean value around 0.0033[44].

Although a linear trend was observed among normal clones, the fixation rate is not constant. In order to investigate the difference in fixation rate among the normal clones, we split all the normal clones into two groups, one group with a fixation rate higher (102 evolution runs) than the mean fixation rate and the other group (436 evolution runs). Results show that the former group is enriched with mutations in genes governing DNA repair, DNA supercoiling or DNA assembling or encoding DNA polymerase, DNA gyrase, DNA glycosylase, DNA repair proteins and DNA-binding proteins (33% vs 17%; $p$ value: 0.00058, Supplementary Data 4). As expected, all strains are not equal, with BW25113 and W3110 having the highest/lowest mutation rates (Supplementary Data 7), while ANOVA analysis indicates that other factors including stress, medium and generation also contribute to the higher/lower mutation rate (Supplementary Fig. 6. b and c; Supplementary Information).

**Mutation-based clustering is informative of the action mechanism.** Based on the mutation profiles, we grouped all the evolution runs which had the same strain and medium, resulting in 4 groups with 22 (antibiotics), 9 (antibiotics), 5 (butanol, osmotic, $H_2O_2$, acidic) and 3 (antibiotics) stresses in each. As expected, the mutation-based clustering has information about the mechanism of action (Supplementary Tables 1 and 2). In Fig. 3d, the dendrogram from the hierarchical clustering of the

**Table 1 The hotspot genes in non-mutator and mutator strains**

| Non-mutator strains | | | | Mutator strains | | | |
|---|---|---|---|---|---|---|---|
| Gene | Function | Frequency | P value | Gene | Function | Frequency | P value |
| rpoB | RNA polymerase sigma 24 | 29.8% | −99 | gyrB | DNA gyrase | 52.9% | −22 |
| pykF | Pyruvate kinase I | 20.2% | −85 | rpoB | RNA polymerase sigma 24 | 41.2% | −18 |
| rph | RNase PH | 15.5% | −47 | chiA | Endochitinase | 41.2% | −19 |
| malT | MalT-maltotriose-ATP DNA-binding transcriptional activator | 14.8% | −42 | rpoC | RNA polymerase sigma 24 | 35.3% | −14 |
| ybaL | Putative transport protein, monovalent cation: proton antiporter-2 (CPA2) family | 14.3% | −38 | bcsZ | Endo-1,4-D-glucanase | 35.3% | −15 |
| rbsK | Ribokinase | 14.3% | −40 | adhE | Aldehyde alcohol dehydrogenase | 35.3% | −13 |
| rbsA | Ribose ABC transporter | 14.3% | −41 | mreC | Membrane protein required for maintenance of rod shape | 29.4% | −12 |
| spoT | Guanosine 3ʳ- diphosphate 5ʳ- triphosphate 3ʳ- diphosphatase | 13.7% | −26 | secA | Protein translocation ATPase | 29.4% | −13 |
| topA | DNA topoisomerase I | 13.1% | −35 | ycaL | Predicted peptidase with chaperone function | 29.4% | −11 |
| nadR | NadR DNA-binding transcriptional repressor and NMN adenylyl transferase | 13.1% | −36 | eptB | Kdo2-lipid A phospho-ethanolamine7-transferase | 29.4% | −10 |
| rbsD | Ribose pyranase | 13.1% | −38 | elfC | Predicted outer membrane usher protein | 29.4% | −10 |
| rbsB | Ribose ABC transporter | 13.1% | −39 | fepE | Obactin (enterochelin) transport | 29.4% | −11 |
| rbsR | RbsR-ribose | 13.1% | −36 | ravA | Regulatory | 29.4% | −9 |
| fis | Fis DNA-binding transcriptional dual regulator | 13.1% | −32 | yeaH | Conserved protein | 29.4% | −11 |
| hslU | HslVU protease | 12.5% | −35 | glnE | Glutamine synthetase adenylyl transferase glutamine synthetase deadenylase | 29.4% | −12 |
| envZ | EnvZ sensory histidine kinase | 12.5% | −35 | rhsC | RhsC protein in rhs element | 29.4% | −9 |
| hsrA | Putative transport protein, major facilitator superfamily (MFS) | 11.9% | −30 | ybjL | Inner membrane protein YbjL | 29.4% | −8 |
| metL | Aspartate kinase/ homoserinede-hydrogenase | 11.3% | −27 | ymfD | e14 prophage; predicted SAM-dependent methyltransferase | 29.4% | −11 |
| mrdB | Rod shape-determining membrane protein; sensitivity and drug | 10.7% | −30 | yrfF | Inner membrane protein inhibits the Rcs signaling pathway | 29.4% | −10 |
| iclR | IclR-glyox | 10.1% | −28 | glyQ | Glycyl-tRNA synthetase | 29.4% | −12 |

The p value is in $\log_{10}$ scale

first group (22 antibiotics) is shown, which follows the mechanism of action, except in the case sulfamethaxozole and sulfamonomethoxine. Interestingly, these two antibiotics also cluster together when mutation frequency is taken into account (Supplementary Fig. 8).

**Evolutionary signatures increase linearly with replicate experiments**. The degree to which convergence and contingency impacts evolutionary outcomes remains a subject of debate in evolutionary biology[45–48]. Both convergence, the notion that independent species will end up with similar adaptations[49,50], and contingency, i.e., that trait evolution is not predetermined but a result of chance with alternative end points being possible[51], are present during evolution. The level of convergence is expected to shed light on the predictability of evolution. To address how predictable evolution is from the perspective of genetic mutations, we evaluated the frequency of mutations within multiple replicates under the same condition. We found a linear relationship between the averaged frequency of all the mutations under a specific condition and the reciprocal of the number of biological replicates: $F = 1.49/N + 0.018$, where $F$ is the averaged frequency and $N$ is the number of replicates (Fig. 4a). Interestingly, the variability of the result decreases with frequency. Our results show that when replicates are few, patterns are more similar, while alternative patterns emerge when the number of replicates increases.

**A pattern of shared mutations across replicates**. We then quantified the level of convergence by two distinct ratios: the

global overlapping ratio that corresponds to the percentage of shared mutations among all clones and the pairwise overlapping ratio, which is the percentage of shared mutations between any two clones, given a specific condition. When counting shared mutations, the difference in the mutation type was not considered. We calculated the two ratios as a function of the number of biological replicates for the 73 different culture conditions in our database that have two or more replicates. As expected, the global overlapping ratio is decreasing proportionally to the inverse of the number of replicates, reaching 3% at experiments with 114 replicates, given by $G = 1.5/N + 0.015$, where $G$ is the global overlap ratio and $N$ is the number of replicates. (Fig. 4b). This trend is in line with the curve in Fig. 4a, as the average frequency of each mutation is expected to be similar to the overlap of a mutation profile with all the profiles in a given condition. The difference between the global overlap ration and the null model, i.e., each mutation present in exactly one replicate (dotted line in Fig. 4a), represents the common mutations shared across replicates under any given condition. Interestingly, however, if we examine each pair of replicates together, their overlap ratio decreased slightly across the whole range of replicates per experiment (Fig. 4c). This pattern holds across medium, strain and stress differences. Concomitantly, we observe higher convergence (pairwise overlap ratio) at adverse conditions, such as comparing minimal vs rich media ($p$ value $< 10^{-4}$) and antibiotic vs no stress ($p$ value $< 10^{-9}$, Fig. 4d).

**Predicting evolution through data integration**. Given the repeatability in mutation targets in an environmental setting,

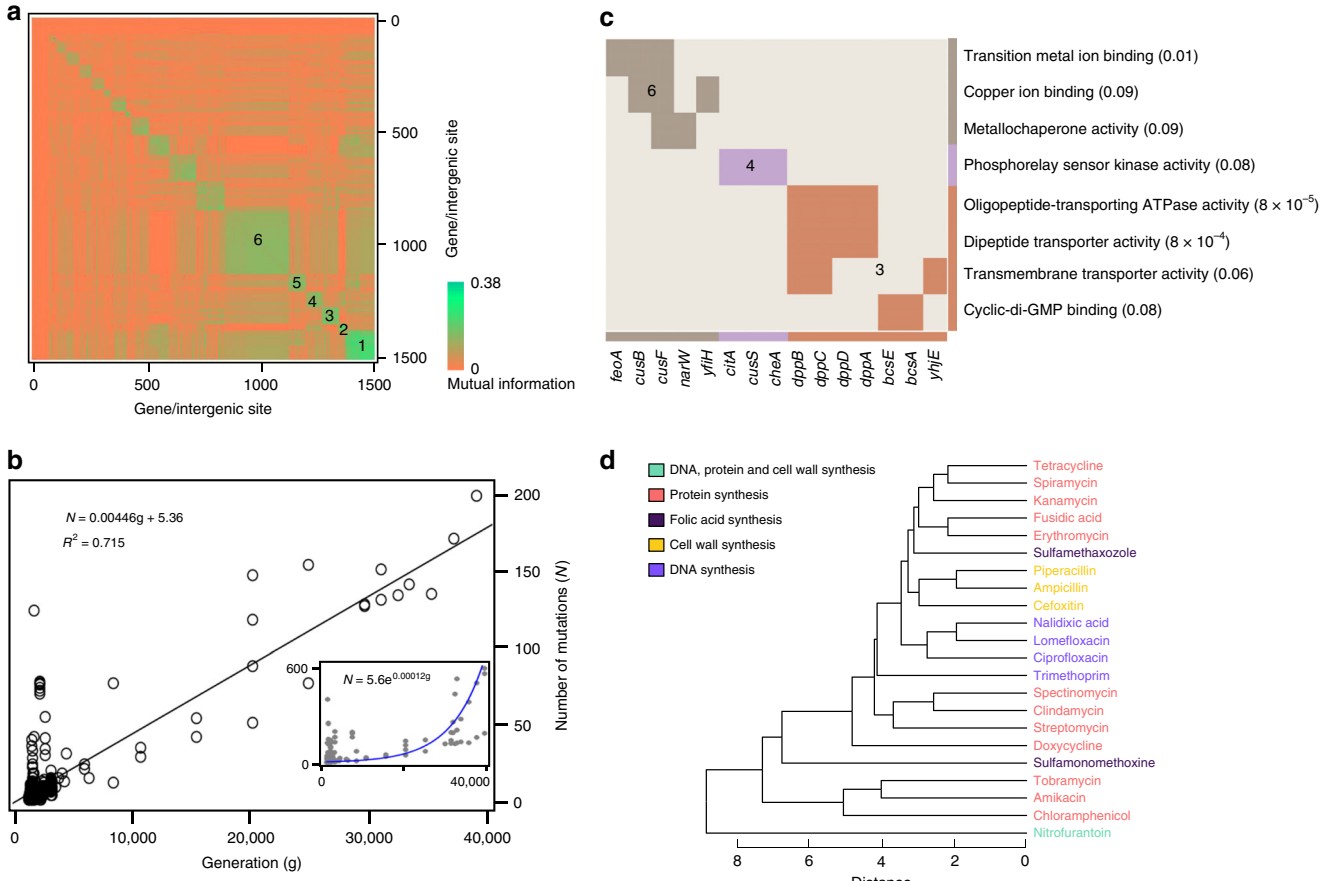

**Fig. 3** Mutation profile analysis for co-occurrence and functional relationships. **a** Spectral clustering of mutations occurring in two or more conditions. Squares along the diagonal represent clusters of genome sites which are highly correlated with respect to mutation profiles. The average pairwise mutual information in each cluster decreases from cluster 1 to cluster 6. Heatmap based on mutual information. **b** The enriched molecular function GO terms and corresponding genes. In each cluster the genes mapped to the same GO terms are together (clusters have different colors and indices). The numbers following each GO term are *p* values calculated by DAVID (*p* value threshold is 0.1). The color bar at the bottom and right indicates the membership of the GO terms/genes among the three clusters. Only clusters with enriched GO terms are shown. **c** The number of mutations as a function of generations elapsed, with hypermutator strains excluded. Inset shows patterns when hypermutators are included with fitted exponential curve in the inner plot being $N = 5.6e^{0.00012g}$. The $R^2$ for linear and exponential fitting is 0.72 and 0.35, respectively. **d** A dendrogram illustrating the clusters of antibiotics generated by hierarchical clustering. The legend describes the action mechanism of each category of antibiotics. When computing the pairwise distance, the Euclidean distance between mutation profiles was used

we used the evolutionary histories to train an Ensemble predictor of which genes are likely to have mutations given a novel environment. In this case, the Ensemble predictor calculates the probability of such event by combining the outputs of an Artificial Neural Network (ANN), a Support Vector Machine (SVM) and a Naive Bayes (NB) classifier. Ensemble predictors have shown to be robust to a wide variety of prediction tasks from the prediction of molecular targets to biological network inference[52]. Here, we use the medium, strain, generation and stress information as features to predict whether a genome site is mutated (Fig. 5a). A bottom-up wrapper method was used for feature selection on all 1990 genome sites (Supplementary Data 8). The top 10 most frequently selected features were found to be antibiotics and temperature. The features related to strain are all less likely to be selected, compared to features related to stress and medium.

As shown in Fig. 5b, c, the prediction performance of each individual predictor was well above the baseline (area under the curve/area under the precision–recall curve (AUC/AUPRC) for the different predictors: ANN, 0.93/0.32; SVM, 0.85/0.13; NB, 0.92/0.18; baseline, 0.69/0.04), with the Ensemble predictor achieving surprisingly high performance under leave-one-condition-out cross-validation (AUC 0.96; AUPRC 0.37). All performance data, confusion matrices and intersection among mutations are included in Supplementary Data 9. To address the class imbalance we used oversampling (see Methods), although the performance of the Ensemble predictor was improved only slightly (Supplementary Table 3). The performance of the Ensemble predictor varies when predicting mutations at different genome sites (Supplementary Fig. 9, AUC: 0.95 ± 0.06, AUPRC: 0.37 ± 0.19). Not surprisingly, the performance is proportional to the frequency of a genome site being mutated across various culture conditions. We investigated how the performance changes for more challenging separations by performing a 10-fold cross-validation, where we found a moderate reduction to AUPRC (0.28 vs 0.37) and almost no change to AUC (0.92 vs. 0.95). The distribution of AUC and AUPRC for the ensemble predictor and each individual predictor is shown in Supplementary Fig. 9c and

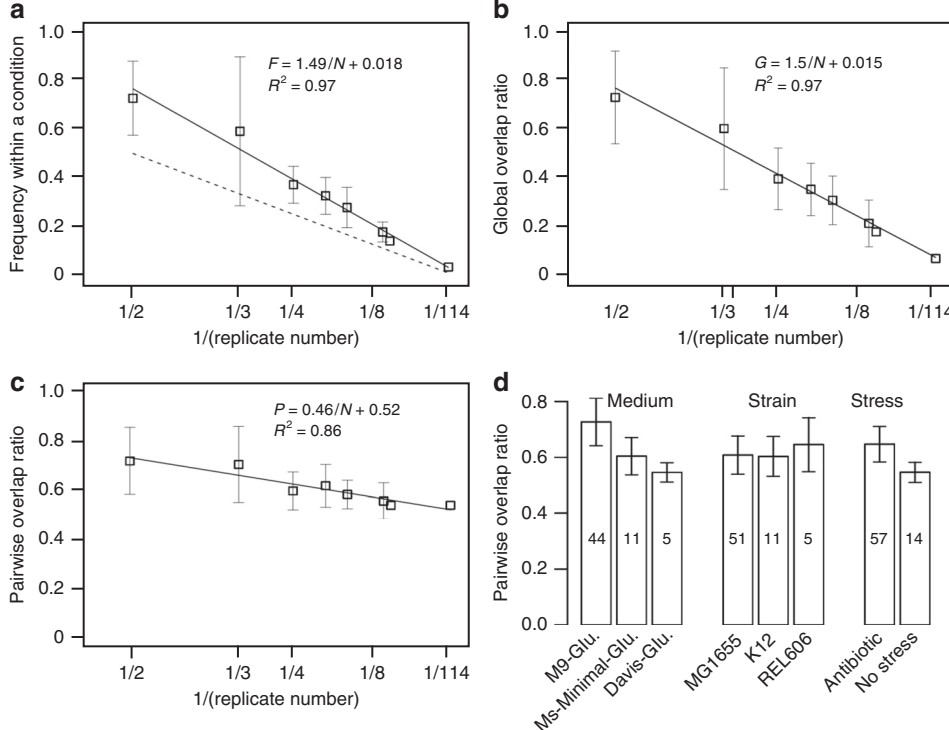

**Fig. 4** Frequency and co-occurrence of mutations as a function of biological replicates. **a** The reciprocal of the number of biological replicates is linearly related to the number of the average frequency of a genome site mutated among all the replicates given in a culture condition. The culture conditions that have the same number of replicates were grouped as one data point in the plot (the dots without standard deviation bar corresponds to the case that only one condition has such number of replicates; this also applies to **b** and **c**). The dash line describes a relation between number of replicates and frequency with a condition if each mutation occurred in only one replicate. **b** The average overlap between the mutated genome sites in a biological replicate and all the mutated genome sites under a culture condition as a function of the number of replicates. **c** The pairwise overlapping in mutated genome sites between biological replicates given a culture condition. **d** The pairwise overlapping ratio under different conditions was addressed against the medium, strain and stress, respectively. The number in each bar represents the number of culture conditions corresponding to that data point. The error bar represents the standard deviation of the pairwise overlapping ratio across the culture conditions corresponding to that data point

9d, with the ANN outperforming other individual predictors (Supplementary Table 4). The intersection of the predictions among the three individual predictors is shown in Supplementary Fig. 10.

To further validate the prediction power of the Ensemble predictor, we applied the method in a novel condition (strain MG1655 in M9 salt media under osmotic NaCl stress). We then performed the experiment by evolving 35 cell lines and then re-sequenced their genomes to identify 23 mutations with varying frequencies (Table 2, Supplementary Data 10). The number of mutations observed in our experiments after 500 generations is close to the average number of mutations in our database, which is 36 for 500 generations. After bootstrapping, the ensemble predictor was able to predict 34.5 ± 5.7% of the mutation targets (recall) at a 49.2 ± 5.8% precision in forward validation (AUC: 0.69 ± 0.08, AUPRC: 0.17 ± 0.03).

## Discussion

In this work, we constructed the first *E. coli* evolution database with a focus on training predictive models of mutation targets. The database enables the user to compare results and obtain statistics, hence addressing important issues in evolutionary biology and extracting knowledge regarding microbial evolution. Most of the mutations acquired during evolution are point mutations (57%) in coding regions (80.3%). Since the *E. coli* genome has about the same ratio of coding regions (e.g., the MG1655 strain has 86% of its DNA in coding regions), the

position of the fixated mutations is not biased towards either intergenic or genic regions. Transcription-related genes such as *rpoB* are among the top genes detected in our study, in agreement with previous work[36]. A single mutation in one of these genes can change cell metabolism or the expression of hundreds of genes with clear fitness implications advantages. Hypermutator phenotypes were attributed to mutations in genes related to DNA replication and DNA repair functions[53] and mismatch repair genes[54]. As expected, the gene *gyrB* which encodes for a subunit of the DNA gyrase is mutated in half of the mutator strains. This is similar to the malfunction in mismatch repair system that was found to contribute to the evolution of cancer cells in mammalian systems[55–59].

In terms of mutation frequency, the most frequent mutation target is *rpoB*, which encodes the β-subunit of bacterial RNA polymerase, mutated in 10 out of the 33 stresses. There are mutations that are specific to the media: for instance, mutation in *satP*, a succinate transporter, is specific to M+Glycerol media and has appeared in 12 out of the 178 respective samples. We measured the number of stresses and media in which each mutation appears (Supplementary Data 11 and Supplementary Fig. 11) and found that although mutations that are common in the majority of stresses/media exist, they are the exception rather than the rule. Indeed, only 1.8 and 21% of the mutations appear in more than 10% of the stresses and media, respectively. These mutations, although more general than others, are associated with specific stresses and media and hence have predictive value as long as the

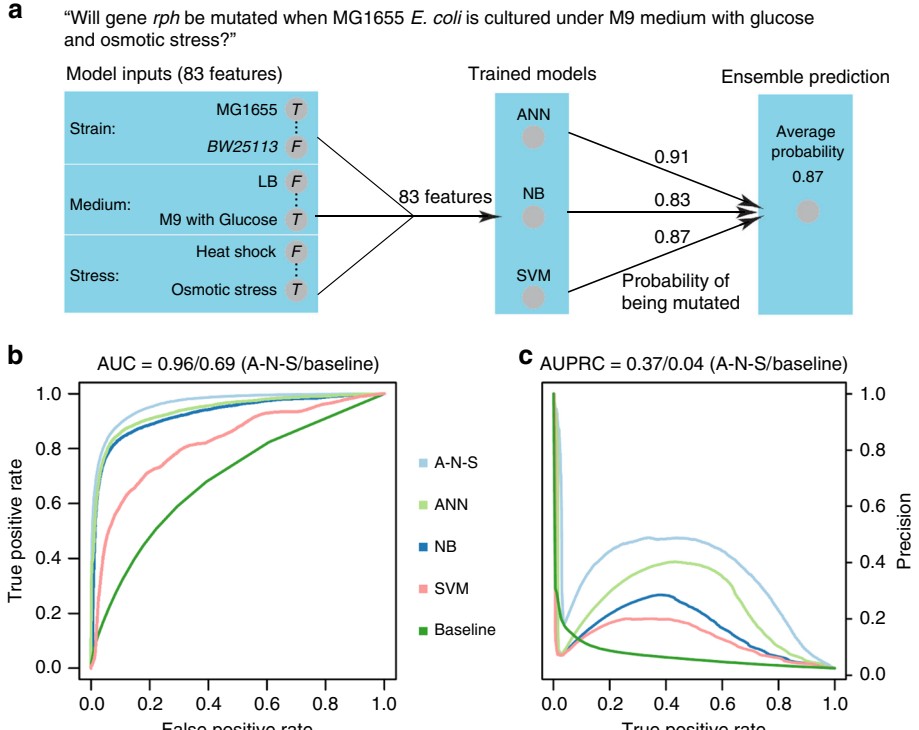

**Fig. 5** Predictor performance. **a** Schematic depiction of the Ensemble predictor and an example of how the probability of a mutation is calculated given an input. The model input consists of 83 binary variables that capture experimental factors. The model output is a binary variable that captures the presence/absence of mutation(s) in a specific gene (*rph* here), given a condition (MG1655, M9 with Glucose and Osmotic stress here). **b** The prediction performance of the ensemble predictor and each individual predictor in leave-one-condition-out-cross-validation. ROC curve. **c** Precision–recall curve. A-N-S is the Ensemble predictor. The baseline refers to the prediction based on the frequency of a mutation across different culture conditions in the database

input embedding to the machine learning methods allows for such associations to take place.

Cross-stress protection is the phenomenon where adaptation in one stress confers a fitness advantage to a second stress[9]. We have found numerous such cases in our analysis, some of them described before and some novel (Supplementary Fig. 5). For instance, the *adhE* gene that was hit in both anaerobic and butanol stresses is known to increase yield in butanol-related fermentation processes. Among the five stresses (heat, anaerobic growth, acid, presence of antibiotics and butanol), we found that mutations linked to anaerobic growth are more prone to also be mutation targets of other stresses. Anaerobic respiration is upregulated at cell repair at the cost of adenosine triphosphate (ATP) production[60], and thus this may be one of the reasons that anaerobic respiration genes tend to be targeted by mutations under other stresses too.

Evolution is a dynamic process, with some significant mutations in terms of fitness advantage not appearing until later in the evolutionary trajectory[61,62]. Due to negative pleiotropic effects, the supply of higher fitness mutations initially is fairly limited[63]. As such, the global overlapping ratio decreases as a function of the number of replicates and the pairwise overlapping stabilizes around 0.53, suggesting an interplay between deterministic and stochastic factors. The degree of convergence reflects the level of repeatability of evolution, which might cast light on its predictability[64]. We noted that presence of antibiotic in the media triggers a remarkably more convergent evolution than the existence of a less dire selection pressures. This can be attributed to the limitation of pathways involved in antibiotic resistance or other environmental limitations[65]. The stress an organism is exposed to creates an imprint on its genome[66], and similar to our

analysis, mutation signatures have been found in ultraviolet light-exposed melanoma[67] and tobacco-exposed and arsenic-exposed[68] lung cancer. Due to less complexity and more target trajectories of microbial populations under adaptive laboratory evolution, this dissection of the mechanistic insights is possible.

We have found that evolution can largely be predicted for conditions that are closely related or are combinations of conditions that are present in the database. We expect the accuracy of the predictions and ability to call out gene targets in environments that are combinations of the strains, media and stresses in the database will continue to increase as we continue to enrich the mutation database. We found that at this stage, an Ensemble predictor that takes integrates different single predictors (Naïve Bayes, Feed-forward Neural networks and Support Vector Machines here) has a superior performance overall with impressive precision and recall, given the task.

There are several areas of improvement. First, new experiments are published each month and a standardization of mutation reported and an automated integration process with the database is important to sustain a consistent mutation registry. In our analysis, we did not take into account a plethora of genomic features (supercoiling states, genetic elements, such as promoters, binding sites, among others) that can be co-analyzed and provide insights into how their presence or absence influence the distribution of mutations. A more extensive investigation of what gene groups (biological processes, molecular functions) are hit under each condition as well as more mechanistic insights can bring more clarity on the molecular basis of these statistical observations. In terms of predictive modeling, there are a number of improvements that are likely to provide superior results as the database size

increases. In the current study, mutations are considered as independent to each other, which is a first-order approximation. Employing techniques that can overcome this limitation, from graphical models to different artificial neural network architectures, are likely to capture these dependencies and result in increased performance. Concurrent mutations are known to be important and dependencies are present in the compendium here (the distribution of the number of dependent genome sites is in Supplementary Fig. 12). Additionally, it would be interesting to investigate whether the order of mutation, captured in time series experiments, can be predicted as well, an investigation that we could no't perform due to the scarcity of time series samples. Another extension would be to predict mutations in environments that have additional dimensions (e.g., presence of more than one species or consortia) at a functional group level considering the interplay between convergence and contingence. Ultimately, methods that integrate statistical analysis similar to the one presented here and mechanistic models have the potential to revolutionize the field as they can predict at a higher resolution (nucleotide level) and provide a causal relationship between the emergence of a mutation and its effect. Until then, the integration of such secondary effects in mixture-of-expert models, such as the one that we constructed here, has the potential to increase our ability to predict evolution and generalize more accurately in novel environments.

## Methods

**Literature curation and DB construction**. A literature-based approach was taken to construct the database (DB). The publications around *E. coli* without whole-genome sequencing was not included due to the lack of a complete screening of possible mutations. The information of experimental setup and mutation spectra was distributed in the body and supplemental material of each publication and reported in different forms. We unified the coding of a culture condition by a binary vector of three categories of attributes: strain, medium and stress. Among the binary variables related to strain and medium, only one can be true and multiple variables related to stress can be true. The duration of each evolutionary experiment was represented by 7 attributes corresponding to 7 time intervals:

(0,500], (500,1000], (1000,5000], (5000,10000], (10000, 20000], (20000, 30000], (30000,40000]. A parenthesis and a square bracket means exclusive and inclusive, respectively.

When constructing the DB, it was noted that the taxonomy for each gene varies in different publications since most genes of *E. coli* have one or more synonyms. To unify the name of all the genome sites hit by a mutation, we replaced all the synonyms for each gene with the gene name adopted in the Ecocyc database. An intergenic region is denoted by the two flanking genes separated by a dash. If an insertion sequence (IS) element is involved in the mutation, the IS element was appended to gene name to distinguish from mutation without an IS element. In some publications only the two flanking genes were reported if a chuck of genome was deleted. In this case, we filled out the in-between genes by looking up to the ancestral genome. Additionally, some evolved strains demonstrated an elevated mutation rate and were classified as mutators in the original publications. The emergence of mutators represented an important and distinct evolutionary behavior from normal adaption evolution in that mutators exhibited advantages over normal evolved strains[69] and are more likely to acquire antibiotic resistance[70]. Thus, evolution runs with a mutator emerging were included and flagged in our database for potential study in the future. In the analysis and prediction in this study, mutator strains were excluded without further explanation except when we compared the mutation spectra of mutators and normal evolutionary strains.

**Evaluating the statistical significance of the hotspot genes**. When calculating the $p$ value for an observed frequency given a gene, the null hypothesis was that the genome is subject to a mutation with equal chance on any sites if the selective factor does not play a role. First, under this hypothesis, the probability of a gene bearing a mutation was computed according to its length. During an evolution run where multiple genes were mutated, the probability of a particular gene being hit was modeled using a binomial distribution. The probability of a gene being hit by a mutation once within multiple evolution runs was modeled by a hypergeometric distribution. Finally, we computed the $p$ value of observing $k$ times of mutation to a particular gene within all the normal/mutator conditions by sampling 100,000 combinations and calculating the average.

**Specifications of spectral clustering**. Spectral clustering approach was applied to detect gene clusters during evolution. First, all the evolution runs that used the same strain, medium and stress were merged, resulting in a binary vector indicating which genome site was hit by a mutation given such an experimental setting. Mutations hitting different positions of a gene were treated with no discrimination. Then, the pairwise correlation between different genome sites was evaluated by the mutual information between every two mutations. Finally, spectral clustering analysis was conducted on the mutual information matrix. The number of clusters is a hyperparameter of spectral clustering algorithm. We tested different

---

### Table 2 Mutations discovered in 35 *E. coli* isolates under osmotic pressure

| Gene | Mutation type | Frequency | Product |
|---|---|---|---|
| **fepA-fes** | SNP | 34 | |
| yjcO-gltP | Insertion | 34 | |
| **rph** | Deletion | 27 | RNase PH |
| pyrE-rph | Deletion | 13 | |
| sufB | SNP | 4 | SufBCD Fe-S cluster scaffold complex |
| **pykF** | SNP | 3 | Pyruvate kinase I |
| yeaJ | Insertion | 3 | Predicted diguanylate cyclase |
| sufS | Deletion | 3 | L-cysteine desulfurase |
| prok | SNP | 2 | Glycine betaine/proline ABC transporter |
| **rpoB** | SNP | 2 | RNA polymerase sigma 24 |
| sufD | SNP | 2 | SufBC2D Fe-S cluster scaffold complex |
| spoT | SNP | 1 | Guanosine 3-diphosphate 5-triphosphate 3-diphosphatase (multifunctional) |
| nusA | Deletion | 1 | Transcription termination/antitermination L factor |
| hns-tdk | Insertion | 1 | |
| pykF-ydhZ | SNP | 1 | |
| acpP-fabG | SNP | 1 | |
| ybbD | Deletion | 1 | Predicted protein |
| ybbG | SNP | 1 | Mechanosensitive channel of miniconductance YbdG |
| trg | SNP | 1 | Chemotaxis signaling complex–ribose/galactose/glucose sensing |
| sufC | SNP | 1 | SufBC2D Fe-S cluster scaffold complex |
| ydjJ | SNP | 1 | Predicted oxidoreductase, Zn-dependent and NAD(P)-binding |
| prfB | SNP | 1 | Peptide chain release factor RF2 |
| nikA | SNP | 1 | Nickel ABC transporter |

The product of each mutated gene was annotated according to http://ecocyc.org/; the product of intergenic region was left blank. The bold genome sites are predicted to be mutated for more than 5 times among 10 times of bootstrapping

hyperparameter settings and Fig. 3a was plotted with the hyperparameter of 19. When implementing spectral clustering, we excluded the genome sites which were mutated only under one condition. The GO term enrichment analysis was conducted using DAVID (Database for Annotation, Visualization and Integrated Discovery)[71].

**Clustering stresses according to mutation profiles**. We partitioned the 574 evolution runs into different groups, within which the same strain and medium was used. A binary mutation profile was generated for each stress in the first group by merging the replicates under a unique stress. We clustered the stresses according to mutation profiles using the built in function "hclust" in R. Different criteria were tried and the criterion taken in Fig. 3d is ward.D2.

**Specifications of each individual predictor**. When training the Naive Bayesian model, we applied Laplace smoothing to prevent getting zeros for the parameters of the model. The Support Vector Machine was trained with a grid search of the c and sigma values, which govern the penalty of misclassification in training set and the width of the Gaussian kernel, respectively (the explored values for c and sigma are [0.001, 0.01, 0.1, 1, 10] and [0.001, 0.01, 0.1, 1] respectively. Keras with Tensorflow as the backend was used to train a feedforward neural network with various architectures, from which one optimal one was selected to capture the relationship between a combination of culture condition and genotype and mutation. The architecture was optimized by a three-step procedure for each genome site. First, the activation function for the hidden layer and optimization method for training a neural network were optimized based on recommendation in literature[72] and our experimental results on part of data. Then, the number of nodes in hidden layer and number of layers were optimized by the random search[73]. Finally, the selected activation function and optimization method were optimized again. When training the model, we chose the log likelihood of the parameters given the training dataset as the objective function rather than least square error because the latter objective led to a vanishing gradient. For the optimal setting, tanh activation function was selected for the hidden layer and Adam optimizer with an initial learning rate of 0.01 was used to train an ANN. The optimal number of nodes and layers varies for different genome site. The most common setting among the 32 settings (Supplementary Table 5) is two hidden layers with 57 nodes and 37 nodes in each layer respectively, and dropout rate being 0.4 (such a setting was selected when predicting mutations on 257 out of the 1990 genome sites). The results for optimizing the architecture in step 1 and step 3 are in Supplementary Table 6 and Supplementary Table 7, respectively.

**Feature selection for each individual predictor**. In order to select the optimal feature subset from the 83 attributes that represent a culture condition and genotype, we applied a stepwise backward feature selection approach. Specifically, we started with all the features and iteratively removed the feature, which led to the best improvement in classification performance, one by one. The iteration was terminated when removal of a feature from the left subset would decrease the performance. The prediction performance was quantified by the area under the receiver operator characteristic (ROC) curve.

**Ensemble predictor**. Previous empirical and theoretical studies have demonstrated that ensemble learning approach yielded more accurate and robust prediction results[52]. This approach is suitable for tackling biological problems because an ensemble composed of a set of models captures multiple aspects of the biological problem of interest, whereas it is often challenging to develop one single model to account for all the aspects of the biological problem. We constructed an ensemble of three machine learning models, an ANN, a SVM and a NB model, to predict the occurrence of mutation to a gene given a culture condition and a starting genotype. The three models vary in complexity. ANN and SVM are discriminative models, whereas NB classifier is a generative model. They have been successfully applied in various contexts. The final prediction, the probability of the occurrence, is an average of the predictions of the three models with equal weight for each model. Since not all attributes defining a culture condition and genotype were equally informative, we conducted feature selection when building each individual model. For each genome site harboring a mutation under more than two conditions, we build an ensemble predictor without considering additional mutation details. The prediction performance of the ensemble predictor was evaluated using a leave-one-condition-out cross-validation. To tackle the variation in the number of replicates for different culture conditions, all the replicates for one culture condition was merged to one binary mutation profile. When cross-validation was conducted, a condition but not a replicate was left out as testing. Since the ROC curve can be optimistic on the performance if the dataset is highly skewed, a precision–recall curve was used to assess the classification performance. The baseline for evaluating the performance of the ensemble predictor is based on the frequency of a mutation across different culture conditions in the database. The minority class (mutation target) in the dataset is heavily skewed and to address the class imbalance we used oversampling, while we have evaluated other techniques such as ADASYN[74] and SMOTE[75] with similar results.

**Experimental evolution and mutation validation**. E. coli MG1655 was used for the laboratory evolution. Adaptive evolution was performed by daily serial dilutions in Minimal M9 medium with 0.3 M NaCl and 0.4% glucose as carbon source at 37 °C. Every 24 h, growing bacteria of 35 independent lines were diluted 1:500 in fresh medium yielding ~7–9 generations per day. To ensure the count of generation elapsed, bacteria were plated on LB agar. Evolution was continued for a total of 500 generations. DNA of the selected clones was extracted using Wizard Genomic DNA Purification Kit (Promega) and sequenced as described in Supplementary Methods (Supplementary table 8).

## Data availability

The E. coli mutation database (MutationDB), code and predictive models are available at http://www.mutationdb.com. The DNA sequencing data of the lab evolution experiments are in Sequence Read Archive under the ID SRP149905.

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

## Acknowledgements

We thank Muntaha Samad and Ameen Eetemadi for their help in setting up the MutationDB.

## Author contributions

V.Z. compiled data from literature, produced the database and conducted the validation experiment. X.W. proofread the mutation database, conducted the analysis, built the ensemble predictor and built the online database. A.T. helped with the mutation analysis from literature. M.K. conducted the bioinformatic part of the WGS for the experimental validation section. X.W., V.Z. and I.T .prepared the manuscript. I.T. conceived the study and supervised all aspects of the project.

## Additional information

**Competing interests:** The authors declare no competing interests.

