## [Peer Review File · Nature Communications]

Reviewer #1 (Remarks to the Author):

Major Revision

1. The title and introduction describe the primary significance of this work as the given framework for predicting mutations. This implies that the manuscript will present substantial content on predicting evolution events using data provided by current literature. Lines 114 through 367 (253 lines) describe the manuscript's results, including 2 table and 4 figures. Lines 277 through 317 (40 lines) describe the results of predicting evolution events, with 1 table and 1 figure. The content explicitly addressing the prediction of evolution events is $40/253=15\%$ of the results, $\frac{1}{2}$ of the tables, and $\frac{1}{4}$ of the figures. The remaining content (85% of the results, etc.) explores trends from the mutation data mining, though how it relates to the manuscript's work in predicting evolution is not made explicit. The manuscript should better describe how the data mining results motivate the pursuit of predicting evolution.
2. The impact of predicting genetic mutation targets in adaptive evolution is not made clear by the manuscript. The manuscript's only content on the motivation for its work are brief references to similar applications of machine learning (lines 82-85). Lacking a clear impact, the manuscript only describes an exercise in consolidating data, mining for trends, applying supervised learning tools, and validating their performance with experimental results.
3. The ensemble can't predict completely novel genetic mutation targets, only those that are present in the training data. Reducing the ensemble's power of novel prediction are the regulatory system mutations that serve as general adaptations across many different stresses and mediums (PMID: 27135538). These regulatory system mutations are found in the ensemble's training data. The manuscript should clarify how and what context this level of prediction power would be impactful.
4. The manuscript didn't explicitly consider normalizing their mutation data relative to the number of replicates in experiments. The set of consolidated experiments has a replicate range of 1 to 115. Without normalization, the genetic mutation trends and predictor performance presented in the results are strongly biased towards experiments with many replicates. The work described in the manuscript should include normalization for the size of experiments to remove the bias in results.
5. We calculated a different precision (ours=50%, theirs=44%) and recall (ours=36%, theirs=33%) for the performance of the ensemble's predictions according to the true and false positives given by in the "ML Forward Mutation Prediction" spreadsheet of supplementary file 9. Authors should ensure that their calculations are correct and that their results and approaches are clear. The prediction performance is a key result in the manuscript's intended contribution; this discrepancy is a critical issue to address before publication.
6. Two genetic regions included in the results (ybcS and proV of Table 1) aren't included in supplemental files or the online mutationdb.com data describing the mutated genetic region data used to train the ensemble. The manuscript's ensemble cannot predict the mutation of a genetic region without including its mutation in the training data. The authors need to ensure that the submitted results data accurately reflect the claims made in the manuscript.
7. The manuscript should demonstrate in more detail that their ensemble's prediction performance is better than that of a naive approach, such as choosing the top 5 most mutated genetic regions. The current discrepancy in the submitted and claimed results don't allow for this verification by reviewers. The ROC and Precision-Recall curves of figure 5 or S5 only illustrate trends and do not demonstrate performance in a manner that can be validated. Additional supplementary files describing the prediction output that generated the ROC and Precision-Recall are necessary to adequately review the ensemble's performance.

8. Line 119: The claim of "For each mutation event, we recorded its genome position" isn't supported by the results or supplementary data: no genomic nucleotide positions are included with any of the mutation data from the manuscript or mutationdb.com.

9. The result descriptions often refer to "mutations", though the abstract majority of supplemental results report on gene level mutation details. This usage of "mutations" is misleading since readers could interpret the results as being generated using additional mutation details.

10. The claim "Gene ontology enrichment showed that 12 out of the top 20 genes most likely to be hit by a mutation are involved in carbohydrate transport and metabolism, an adaptation to the carbon source where the cells grow" (line 136) isn't represented by any results submitted with the manuscript.

11. The functional analysis of "depletion spots" are given, the genes contained within depletion spots are not given in the manuscript's results or supplementary data. Only those genes with mutations have been reported (supplementary file 3).

12. Line 167 The claim "We observe a higher likelihood to hit DNA-related functions in hypermutators" should be quantified.

13. The genetic mutation target results are in binary form, though the ensemble returns probabilities. The manuscript does not describe a means for transforming the ensemble's probability results into binary results.

14. Line 255 and 256: Are the "shared mutations" described by overlapping genes and mutation type (supplemental file 1) or only genes?

15. Figure 4C includes a data point with 114 replicates and a ~60% pairwise overlap ratio. This data point is uniquely described by the 114 replicates from the Tenaillon 2012 ALE (PMID: 22282810). The paper for this experiment had executed the same analysis, though their mutated gene pairwise overlapping ratio was 20%. The manuscript's results shouldn't differ.

16. The integrity of line 260's claim is affected by this discrepancy: "Interestingly, however, if we examine each pair of replicates together, their overlap ratio fluctuates around 60% across the whole range of replicates per experiment (Fig. 4C)."

17. Describe why an Artificial Neural Network, Support Vector Machine, and a Naive Bayes classifier were specifically chosen for the ensemble.

18. Stratification according to features should be used in the cross-validation of the predictor. The use of leave-one-out cross-validation is not appropriate for this data set since it contains groups of relatively homogenous mutation results, as presented in Figure 4C, and will result in overfitting during validation.

19. The forward validation evolution conditions (duration of evolution, passage volume) and mutation calling toolset should be included, otherwise, the experiment can't be reproduced.

20. Line 324: "80.3%" isn't clearly associated with any result in the manuscript.

Minor Revision

1. Over 50 spelling, grammar, and formatting errors were encountered. The manuscript requires thorough editing to remove such issues that degrade the clarity of the work.

2. Figure 3C should describe the trendline approaches and fit metrics
3. Figure 3D should describe the dendrogram distance units.
4. Line 85-86 "change to be mutation targets" doesn't have a clear context.
5. The abstract should report the recall performance of the ensemble. Precision isn't adequate on its own when describing single value representations of supervised learning performance.
6. The text refers to "Fig. 1A", "Fig. 1B", and "Fig. 1C", though figure 1 doesn't include subpanels identified as A, B, or C.
7. The orange line of figure 2 D and E need to be described.
8. Figure 2's description contains multiple formatting issues and subpanel D doesn't have a description.
9. The claim " One sole mutation in the transcription machinery can allow the change of expression of full metabolic pathways." needs a reference.
10. To be more clear, Figure 2E should include some form of "We used a 5kb sliding window along the E. coli MG1655 genome to find regions that were most or least likely to be hit by a mutation."
11. The statistical approach described by leveraging a Gamma distribution (line 150) isn't clearly associated with any results. Reasons for choosing the Gamma distribution should be included or reference.
12. The term "coldspots" should be removed if only to be used once.
13. "Depletion spots" should be shown in Fig. 2E.
14. The manuscript should state the reason and references for the following choice and claim (line 158) "...hypermutator strains, strains with a mutation rate larger than 0.1 mutations per genome duplication that have been identified as such. Hypermutators are generally expected to follow a different trajectory during evolution"
15. The method for generating p-values in Table 1 should be described or referenced.
16. line 175: "p value ≤ 0.1 " has no context.
17. Line 176: The clusters 2, 3, 4, and 6 referred to are ambiguous: no clusters 4 and 6 in Fig3B and no cluster 3 in FigS1.
18. Line 178: Claim "The only pathway enriched in any of the clusters are the two-component systems, which is present in 9 genes in cluster 1 and 8 in cluster 4." needs to reference the supporting results.
19. Line 183: Claim "Mutation on these genes is a general response of the cells to improve the acquisition of the carbon and to cancel the ability of biofilm formation." needs to reference supporting material.
20. Line 199: The choice of a 10% p-value for DAVID needs to be clarified.
21. Line 199: DAVID needs a reference.

22.Line 239: No explanation is given for the result " These two exceptions fall in one cluster when replicates are merged by counting mutation frequency (Supplementary Fig. S4).

23.What is the difference between the red and black data points in figure 4A?

24.Line 283: Remove the reference to "biological network inference" since its connection to this work isn't explained or obvious.

25.Figure 5A's workflow does not clearly represent the prediction processes. Does the training data contain 83 different types of each feature or a sum of 83 different types across all features? When is the prediction input introduced?

26.Table 2 contains duplicate entries for sufD and spoT.

27.Line 324: "MG1655 genome harbors 86% of coding regions" needs a reference.

28.The choice of model distributions in the methods section "Evaluating the statistical significance of the hotspot genes" should reference material validating these choices.

29.Line 326: rpoB/C/S are directly involved in "transcription" and not "translation".

30.Line 326: The claim that rpoS is a top mutated gene is incorrect according to Table 1, which doesn't include rpoS.

31.Line 351: "limitation of pathways involved in antibiotic resistance" needs a reference.

32.Line 360: "the ensemble predictor was quite accurate". Accuracy was never used as a metric of prediction performance.

33.Line 375: "four categories of attributes: strain, media, and stress." Only 3 categories given.

34.Line 375: "The duration of each evolutionary experiment was represented by 7 attributes corresponding to 7 time intervals." Nowhere in the mutation data is ALE duration described by 7 time intervals.

Reviewer #2 (Remarks to the Author):

This paper describes a meta-study of E.coli evolution experiments, in which mutation data under different conditions of stress and adaptive evolution are harvested for a predictive framework based on machine learning methods. The novelty and strong point of this paper is the combination of data from different experiments into a unifying analytical framework, which drastically increases the information about repeatable mutation patterns and lays the ground for training predictive methods. At the same time, this strategy is what raised most of my questions on the paper, which are on the justification of pooling data for specific inference purposes and on the validity of some results. I think the points below should be carefully addressed in a revision.

1. Increase of substitution rate with time. This point, shown in fig. 3c, puzzled me most and I have difficulties in rationalising it. Most adaptive evolution experiments can be expected to take place under clonal interference, where the rate of substitutions is nearly constant and their fitness effects are a slowly decreasing function of time (this is the case, for example, in the Lenski experiment). To analyse the pattern observed here: Is the same pattern observed for synonymous and non synonymous changes? For the long term experiments, there should be data at intermediate time points; do they follow the same pattern? I am worried that the long-term

experiments may target systematically other strains or selective stresses, which can confound the results.

2. Repeatability of mutational patterns. The pattern observed in Fig. 4a-c can probably be digested in a more informative probabilistic model giving the likelihood to mutate a given gene in a given condition, such that the enhancement of repeatability over a null model of independent mutations (no repeatability) becomes clear. For example, a simple null model of independent mutations would give a frequency of $f = 1/k$ for a given mutation under k replicate experiments. The pattern of fig. 4a is a bit above that, $f \sim 1.5 / k$; how does that relate to the pattern of fig. 4b can 4c?

3. Inference of epistasis. The analysis of co-occurrence and epistasis on p.8. appears to have an inconsistency. Epistasis means that the selection on one mutation depends on the genetic background it appears on, so only subsequent mutations in the same evolving population can be counted and the data must not be pooled across populations (cf. Methods, line 403-404). I realise that the data may not be informative enough under this more stringent requirement; the analysis of epistasis is not central to the paper and could also be left out without too much loss.

4. Predictive methods. The discussion of this central point of the paper is too short and too much of a black box. It would be interesting to know the consistency of predictions from the different methods. A naive look at fig. 5b suggests that the other methods do not add much to a naive Bayes prediction, perhaps the main analysis should be limited to that method? What happens if more challenging separations into training and validation data are applied? Can we learn informative and biologically meaningful clusters of features, i.e., of strains, media, and stress conditions? Key features of the prediction methods should be described in Methods and Supporting Information to make the paper reasonably self-contained for a non-specialist readership.

4. Minor points: (a) Figs. 2cd would be more convincing on a log-log scale. (b) The discussion can be improved in its coherence. For example, the second and third paragraph on p. 16 list a number of statements on evolution; I am not sure if all of them reflect the intention of the papers cited and how these statements are linked to the central results of the paper. A critical discussion of the state of predictions, including the current level of accuracy and ways for improvement, would be more appropriate.

Reviewer #3 (Remarks to the Author):

This manuscript describes the significant effort undertaken by Wang et al. to survey, compile and curate E. Coli mutation events under different environmental settings. Except for a few minor errors, I found the paper well written with a logical flow and interesting findings. The major contribution of this paper is an "ensemble mutation predictor" that predicts which gene/intergenic regions are going to be mutated under a certain condition. I have a few questions and I think they might help me and the readers understand the method better:

1. The authors mention that there are 83 features and they are shown as binary (T/F) individual features in Fig. 5A. (This pane is a bit confusing and might imply that there are 3 sets of features.) It seems that a couple of these features are categorical (e.g., Strain, Medium and Time) and should be encoded as 1-of-K. That is, only one Strain feature can be True and all other Strain features must be False. Stress seems to be different though, as more than one Stress feature can be True. I think feature encoding should be clearly defined and the number of features be updated (all Strain features count as one). Moreover, feature selection should be done differently on categorical features (all levels must be treated jointly).

2. If I understand correctly, there are 1,990 models for 1,990 gene/intergenic regions and all these models are trained independently, whereas mutations observed are presumably dependent. What would be the rationale for assuming independence and how this assumption can be

verified/tested?

3. Since there are 1,990 models and 1,990 sets of performance metrics, reporting metrics must include their statistical variation. For example, in Fig. 5B, the ROC curves are averages over all models. Which one is the best-performing model and which one is the worst-performing model? What is the variation in AUC and AUPRC (median, SD, etc)? (On a side note, the title of Fig. 5C is incorrect). I think boxplots of AUCs and AUPRCs could be informative. Moreover, a statistical test must be performed to show superiority of one model over another. In Fig. 5C, the difference between ANS and all underlying models is huge. What is the explanation? Also, I do not think PRCs can be averaged.

4. The frequency of mutation at each region is very low and the training problem is extremely imbalanced. How is this incorporated in the training process? I would recommend bootstrap aggregating (bagging) combined with oversampling of the rare class.

5. Why does the Naïve Bayes model perform so much better than the other two very flexible models? The ANN model seems very simple (only 6 nodes per layer). Are modern neural net frameworks such as TensorFlow or PyTorch are being used?

6. The ROC curve in Fig. S5A is not accurate. In general, when there are few data points or there are ties in the predicted scores, one must bootstrap to report uncertainty. That is, sample n points with replacement, compute AUC and AUPRC, repeat 10 or 100 times and report mean AUC and AUPRC and their SDs.

We would like to thank the reviewers for careful and thorough reading of our work and for the thoughtful comments and constructive suggestions. We have responded each point brought by the reviewers as follows (Each response follows the corresponding remark by the reviewers; remarks are in italics)

1. The title and introduction describe the primary significance of this work as the given framework for predicting mutations. This implies that the manuscript will present substantial content on predicting evolution events using data provided by current literature. Lines 114 through 367 (253 lines) describe the manuscript's results, including 2 table and 4 figures. Lines 277 through 317 (40 lines) describe the results of predicting evolution events, with 1 table and 1 figure. The content explicitly addressing the prediction of evolution events is 40/253=15% of the results, ½ of the tables, and ¼ of the figures. The remaining content (85% of the results, etc.) explores trends from the mutation data mining, though how it relates to the manuscript's work in predicting evolution is not made explicit. The manuscript should better describe how the data mining results motivate the pursuit of predicting evolution.

Response: We agree that a large portion of the manuscript is on analysis of the dataset and trends within and this has been done so by design. We believe understanding the data is paramount to any successful prediction effort, as it is quite easy to be misguided otherwise due to data noise, biases and structure. While our goal is to investigate the degree that evolution is predictable, the road to Ithaca is as important. For example, the way we defined conditions (lines 88-89) is part of feature engineering necessary in the machine learning section of the paper, despite being introduced earlier. The hypothesis that the same targets are hit during independent evolution experiments in the same environments, hence mutations can be used as features in predictors, is supported in the section about evolutionary convergence (lines 185-214). Similarly, we explain the predicted mutations (machine learning validation) in a forward validation experiment only under the light of the dataset presented earlier on. To clarify these points, we modified the manuscript in lines 189-190, 216-217, 251-252.

2. The impact of predicting genetic mutation targets in adaptive evolution is not made clear by the manuscript. The manuscript's only content on the motivation for its work are brief references to similar applications of machine learning (lines 82-85). Lacking a clear impact, the manuscript only describes an exercise in consolidating data, mining for trends, applying supervised learning tools, and validating their performance with experimental results.

Response: We have now expanded on how predicting evolution can affect our understanding of the interplay between environment and evolutionary processes, as well as lead to better experimental designs for testing hypotheses (lines 65-71).

3. The ensemble can't predict completely novel genetic mutation targets, only those that are present in the training data. Reducing the ensemble's power of novel prediction are the regulatory system mutations that serve as general adaptations across many different stresses and mediums (PMID: 27135538). These regulatory system mutations are found in the ensemble's training data. The manuscript should clarify how and what context this level of prediction power would be impactful.

Response: We agree with the reviewer, statistical learning methods can only generalize past histories to novel input combinations that share fundamental attributes. If we encounter "regulatory system mutations" that function as general adaptations, those would potentially be present ubiquitously in our database, and in that case a solution would be to filter them out during data preparation. However, this is not the case in Mutation DB, as no mutation is present in all conditions. The most frequent mutation target is *rpoB*, which encodes the β subunit of bacterial RNA polymerase, mutated in 10 out of the 33 stresses. Instead, there are mutations that are specific to media: for instance, mutation in *satP*, a succinate transporter, is specific to M9 + Glycerol media and has appeared in 12 out of the 178 respective samples. To address this question, we measured the number of stresses and media that each mutation appears, please find the detailed analysis with Supplementary File 11 and the corresponding histogram in Supplementary Figure

10. Mutations that are common in the majority of stresses/media exist, but they are the exception rather than the rule. Indeed, only 1.8% and 21% of the mutation sites appear in more than 10% of the stresses and media, respectively in our database. These mutations, although more general than others, are associated with specific stresses and media and hence have predictive value as long as the input embedding to the machine learning methods allow for such associations to take place. This is the case here and we clarify this point in lines 262 to 271, where we also cite the relevant publications.

4. *The manuscript didn't explicitly consider normalizing their mutation data relative to the number of replicates in experiments. The set of consolidated experiments has a replicate range of 1 to 115. Without normalization, the genetic mutation trends and predictor performance presented in the results are strongly biased towards experiments with many replicates. The work described in the manuscript should include normalization for the size of experiments to remove the bias in results.*

Response: We have added normalization and all the replicates for a given culture condition were merged as a binary mutation profile to normalize the variation in the number of replicates for different conditions. Leave-one-condition-out cross validation was used when testing the performance of the prediction. We have elucidated this in lines 416-418.

5. *We calculated a different precision (ours=50%, theirs=44%) and recall (ours=36%, theirs=33%) for the performance of the ensemble's predictions according to the true and false positives given by in the "ML Forward Mutation Prediction" spreadsheet of supplementary file 9. Authors should ensure that their calculations are correct and that their results and approaches are clear. The prediction performance is a key result in the manuscript's intended contribution; this discrepancy is a critical issue to address before publication.*

Response: Indeed, we verify the recall of 36% (discrepancy appeared from one gene appearing twice being re-counted in table 2). We has corrected that in the manuscript and the supplemental material. Among the 9 predicted mutations, 4 were really mutated in the experiments, which gives a precision of 44% (4/9). As another reviewer suggested, we now also use bootstrapping to generate multiple datasets to test the robustness of our ensemble predictor, we have included the mean and standard deviation for precision and recall when reporting the results in the abstract and body of the manuscript (lines 13-15, 246).

6. *Two genetic regions included in the results (ybcS and proV of Table 1) aren't included in supplemental files or the online mutationdb.com data describing the mutated genetic region data used to train the ensemble. The manuscript's ensemble cannot predict the mutation of a genetic region without including its mutation in the training data. The authors need to ensure that the submitted results data accurately reflect the claims made in the manuscript.*

Response: We rechecked the genetic regions in Table 1 and all the genes listed in Table 1 are included in the database. We searched for "ybcS" across the manuscript and did not find this entry. The gene "proV" was in Table 2 and it corresponds to "proK" in the database, which now is consistent between Table 2 and DB.

7. *The manuscript should demonstrate in more detail that their ensemble's prediction performance is better than that of a naive approach, such as choosing the top 5 most mutated genetic regions. The current discrepancy in the submitted and claimed results don't allow for this verification by reviewers. The ROC and Precision-Recall curves of figure 5 or S5 only illustrate trends and do not demonstrate performance in a manner that can be validated. Additional supplementary files describing the prediction output that generated the ROC and Precision-Recall are necessary to adequately review the ensemble's performance.*

Response: ROC and PR curves, along with AUCs are provided in Figure 5, with clear evidence that the Ensemble predictor is a superior technique. We believe that taking the top 5 most mutated regions

although reasonable, it would not be a better performance comparison (why 5 and not 10 or 15? Who sets that arbitrary threshold?). However, the prediction output of all individual predictors and the Ensemble predictor is included in Suppl. File 9 and we have now also included a confusion matrix to clarify the calculation of those statistical measures. The baseline we used is the frequency of a mutation across all the culture conditions in the database (lines 420-421), which is the also approach the reviewer suggested.

8. Line 119: *The claim of “For each mutation event, we recorded its genome position” isn’t supported by the results or supplementary data: no genomic nucleotide positions are included with any of the mutation data from the manuscript or mutationdb.com.*

Response: We have added the location of a mutation when reporting the type of mutation in the Supplementary File 2.xlsx.

9. *The result descriptions often refer to “mutations”, though the abstract majority of supplemental results report on gene level mutation details. This usage of “mutations” is misleading since readers could interpret the results as being generated using additional mutation details.*

Response: All 83 features that have been used for prediction are provided in Suppl. File S8 and the methods section has a detailed explanation on what features were taken into account. In our analysis “mutations” means two or more genome sites have one or more nucleotide change. In the manuscript (abstract and results) we note that the predictions are at “gene level” and we now add it in the introduction too to ensure that predictions are made at gene and not nucleotide level (lines 76-77): “Then we used it to train “evolution” predictors that have the capacity of predict gene mutation targets, at gene (not nucleotide) level, given a novel environmental setting”

10. *The claim “Gene ontology enrichment showed that 12 out of the top 20 genes most likely to be hit by a mutation are involved in carbohydrate transport and metabolism, an adaptation to the carbon source where the cells grow” (line 136) isn’t represented by any results submitted with the manuscript.*

Response: We now have added a graph depicting the gene ontology enrichment for the top 20 genes (Supplementary Material, Fig. S1).

11. *The functional analysis of “depletion spots” are given, the genes contained within depletion spots are not given in the manuscript’s results or supplementary data. Only those genes with mutations have been reported (supplementary file 3).*

Response: A new sheet describing the depletion spots was added to the Supplementary File 3.xlsx (also please see legend in the file). In addition, a Figure summarizing the enrichment results was added to Supplementary Material (Suppl. Fig. 3). We also added clarification about depletion spots in lines 112-114.

12. *Line 167 The claim “We observe a higher likelihood to hit DNA-related functions in hypermutators” should be quantified.*

Response: Among the 36 hypermutators, 100% have at least one DNA repair related gene mutated. In contrast, the percentage is 51% for non-hypermutator lines. We now have added the following clarification in lines 128-130 and we list the DNA related genes in Supplementary File 4.

13. *The genetic mutation target results are in binary form, though the ensemble returns probabilities. The manuscript does not describe a means for transforming the ensemble’s probability results into binary*

results.

Response: We set a threshold to transform the predicted probability into a binary result. As the threshold varied, we got a ROC curve and a precision-recall curve (clarified in lines 418-422).

14. Line 255 and 256: Are the “shared mutations” described by overlapping genes and mutation type (supplemental file 1) or only genes?

Response: Shared mutations are described by overlapping genes only. We have now clarified this point in line 201.

15. Figure 4C includes a data point with 114 replicates and a ~60% pairwise overlap ratio. This data point is uniquely described by the 114 replicates from the Tenaillon 2012 ALE (PMID: 22282810). The paper for this experiment had executed the same analysis, though their mutated gene pairwise overlapping ratio was 20%. The manuscripts results shouldn't differ.

Response: We would like to thank the reviewer for the detailed review. We rechecked our computation and found that the discrepancy is attributed to the difference in the ways to compute the pairwise overlap ratio. In the original paper, it was observed “Among point mutations, none of the 36 synonymous and 157 of the 634 nonsynonymous mutation were shared among two or more lines”. We assume the pairwise overlap ratio the reviewer computed is equal to 23% ($157/(36+634)$). Instead, the way we compute pairwise overlap for two cell lines is different: the overlap ratio is equal to the number of mutations in one cell line divided by the number of unique mutations in both lines. We would like to illustrate the difference between these two ways of calculation in the following example: In the table below, each row represents a replicate and the presence of a cross mark indicates the gene in a column is mutated. If we compute the overlap ratio as in the Tenaillon paper, we get a ratio of 33% (2/6), while with our calculation, we get an overlap ratio of 50% (2/4).

	Gene 1	Gene 2	Gene 3	Gene 4	Gene 5	Gene 6
Replicate 1	X	X	X			
Replicate 2	X	X		X		
Replicate 3	X	X			X	
Replicate 4	X	X				X

In addition, the Tenaillon paper also mentioned that “In contrast to point mutations, 69% (82/119) of larger deletion were identical between at least two lines.” Thus, if both point mutations and deletion mutations are taken together as in our work, the numbers would also be different than the 23%.

16. The integrity of line 260's claim is affected by this discrepancy: “Interestingly, however, if we examine each pair of replicates together, their overlap ratio fluctuates around 60% across the whole range of replicates per experiment (Fig. 4C).”

Response: Addressed as part of response to point 15.

17. Describe why an Artificial Neural Network, Support Vector Machine, and a Naive Bayes classifier were specifically chosen for the ensemble.

Response: For two reasons: First the diversity of the multiple predictors is critical for building an effective ensemble predictor and these three methods vary in complexity. In addition, the ANN and SVMs are

discriminative models whereas the Naive Bayes classifier is a generative model, while these three techniques are the most widely used. We have added this explanation in the method part (lines 409-410).

18. Stratification according to features should be used in the cross-validation of the predictor. The use of leave-one-out cross-validation is not appropriate for this data set since it contains groups of relatively homogenous mutation results, as presented in Figure 4C, and will result in overfitting during validation.

Response: We actually used a leave-one-condition-out (LOCO) cross validation, as described in the Methods, line 425 and used successfully in the past (see Kim et al., 2016; Carrera et al., 2014). We clarified this point in the manuscript as it was sometimes referred as “leave-one-out” CV, which would be misleading and we agree with the reviewer that this would have been inappropriate.

19. The forward validation evolution conditions (duration of evolution, passage volume) and mutation calling toolset should be included, otherwise, the experiment can't be reproduced.

Response: We have now added detailed explanation about the method for mutation calling in the Supplementary Material (Supplementary methods).

20. Line 324: “80.3%” isn't clearly associated with any result in the manuscript.

Response: Clarification added in line 252.

Minor Revision

1. Over 50 spelling, grammar, and formatting errors were encountered. The manuscript requires thorough editing to remove such issues that degrade the clarity of the work.

Response: we have checked spelling, grammar and formatting errors.

2. Figure 3C should describe the trendline approaches and fit metrics

Response: The trend-line has been described in the Fig. 3c and fit metrics were described in the caption.

3. Figure 3D should describe the dendrogram distance units.

Response: The distance is Euclidean distance between two clusters. We have added the description in the caption for Fig. 3d.

4. Line 85-86 “change to be mutation targets” doesn't have a clear context.

Response: We have revised the sentence to “Data from genome-wide association studies have been used in various studies including for calculation of the gene mutation probability³³ and the role of pleiotropy in adaptation” (lines 55-57).

5. The abstract should report the recall performance of the ensemble. Precision isn't adequate on its own when describing single value representations of supervised learning performance.

Response: We have added the recall performance in the abstract (line 15).

6. The text refers to “Fig. 1A”, “Fig. 1B”, and “Fig. 1C”, though figure 1 doesn’t include subpanels identified as A, B, or C.

Response: We have corrected the numbering of the subpanels in Fig. 1 to be consistent with the numbering in the text.

7. The orange line of figure 2 D and E need to be described.

Response: We have added a description in the caption.

8. Figure 2’s description contains multiple formatting issues and subpanel D doesn’t have a description.

Response: We have corrected the issues and added a description for subpanel D in the caption.

9. The claim “One sole mutation in the transcription machinery can allow the change of expression of full metabolic pathways.” needs a reference.

Response: We have added a reference for this claim (reference 51).

10. To be more clear, Figure 2E should include some form of “We used a 5kb sliding window along the *E. coli* MG1655 genome to find regions that were most or least likely to be hit by a mutation.”

Response: We have added this clarification in the caption for Fig 2e.

11. The statistical approach described by leveraging a Gamma distribution (line 150) isn’t clearly associated with any results. Reasons for choosing the Gamma distribution should be included or reference.

Response: The Gamma distribution is suitable for fitting long tailed sample distributions as the one observed here (now clarified in line 112).

12. The term “coldspots” should be removed if only to be used once.

Response: Coldspot is used as an antonym of hotspot, but we removed it to avoid confusion.

13. “Depletion spots” should be shown in Fig. 2E.

Response: We considered highlighting the depletion spots in Fig. 2E. However, we observed 38 depletion spots (each of 5kb) that we did not include them in the figure due to space limitations. We have added the depletion spots in the worksheet of Supplemental File 2 and we also mentioned it in the manuscript (line 118).

14. The manuscript should state the reason and references for the following choice and claim (line 158) “...hypermutator strains, strains with a mutation rate larger than 0.1 mutations per genome duplication that have been identified as such. Hypermutators are generally expected to follow a different trajectory during evolution”

Response: 98.5% of replicates show a mutation rate less than 0.1 mutations per genome and although the definition always is under a given assumption, it is clear that hyper-mutators should be between 1% and 5% of the samples with the highest frequency.

15. The method for generating p-values in Table 1 should be described or referenced.

Response: The method for generating p-values in Table 1 was described in the method part (lines 351-359). We have added reference in the method part (reference 106).

16. line 175: "p value <=0.1" has no context.

Response: The p-value threshold corresponds to the statistical significance cut-off for the biological process GO terms. Clarification added in the manuscript (line 136).

17. Line 176: The clusters 2, 3, 4, and 6 referred to are ambiguous: no clusters 4 and 6 in Fig3B and no cluster 3 in FigS1.

Response: Not all clusters have enriched molecular function GO terms and only clusters with enriched GO terms were included in Fig 3B. The original Fig S1 describes the cellular component GO terms and it has been now renamed to Fig 4 (line 137).

18. Line 178: Claim "The only pathway enriched in any of the clusters are the two-component systems, which is present in 9 genes in cluster 1 and 8 in cluster 4." needs to reference the supporting results.

Response: We added a figure that summarizes the pathway enrichment of the clusters in the Supplementary Material (Fig 4).

19. Line 183: Claim "Mutation on these genes is a general response of the cells to improve the acquisition of the carbon and to cancel the ability of biofilm formation." needs to reference supporting material.

Response: We have added references: reference 57 and 58.

20. Line 199: The choice of a 10% p-value for DAVID needs to be clarified.

Response: This is the default DAVID p-value threshold (reference 57) and it has been used in the past (for example see here (reference 58)). Usually we select the more stringent 0.05 threshold when reduction of the false positives is important and the more relaxed 0.1 threshold when we perform a systems analysis with no follow-up candidates, as it is the case here.

21. Line 199: DAVID needs a reference.

Response: We have added a reference for DAVID (reference 107-108).

22. Line 239: No explanation is given for the result "These two exceptions fall in one cluster when replicates are merged by counting mutation frequency (Supplementary Fig. S4).

Response: We clarified this sentence (line 182-184).

23. What is the difference between the red and black data points in figure 4A?

Response: The red data points have one replicate and no standard deviation bar was included for such points. We have added this annotation in the caption for Fig. 4a.

24.Line 283: Remove the reference to “biological network inference” since its connection to this work isn’t explained or obvious.

Response: The paper referenced used an Ensemble method from 35 other methods and concluded that an Ensemble method is a great generalist that provides superior performance across a wide variety of datasets, hence it is more robust than any single method. The fact that they used the Ensemble technique for network inference, while we do it for mutations doesn’t change its fundamental characteristics (line 220).

25.Figure 5A’s workflow does not clearly represent the prediction processes. Does the training data contain 83 different types of each feature or a sum of 83 different types across all features? When is the prediction input introduced?

Response: The training data contain 83 different types of features. We have modified the Fig 5 a to clarify that. We assume the reviewer refers to the numbers (0.91, 0.83, 0.87) that are above the arrows associated with the “prediction input”. If so, the prediction inputs are generated by three models (ANN, NB, SVM).

26.Table 2 contains duplicate entries for sufD and spoT.

Response: We have removed the duplicate entries in Table 2.

27.Line 324: “MG1655 genome harbors 86% of coding regions” needs a reference.

Response: We have added reference for the GTF file used in this paper (reference 77).

28.The choice of model distributions in the methods section “Evaluating the statistical significance of the hotspot genes” should reference material validating these choices.

Response: We have added reference that substantiates why these methods have been picked and how they are calculated through Monte Carlo (reference 106).

29.Line 326: rpoB/C/S are directly involved in “transcription” and not “translation”.

Response: Corrected.

30.Line 326: The claim that rpoS is a top mutated gene is incorrect according to Table 1, which doesn’t include rpoS.

Response: We have deleted rpoS (line 255)

31.Line 351: “limitation of pathways involved in antibiotic resistance” needs a reference.

Response: We have added the reference (reference 95).

32.Line 360: “the ensemble predictor was quite accurate”. Accuracy was never used as a metric of prediction performance.

Response: Corrected (line 293).

33.Line 375: “four categories of attributes: strain, media, and stress.” Only 3 categories given.

Response: Corrected (line 332).

34.Line 375: “The duration of each evolutionary experiment was represented by 7 attributes corresponding to 7 time intervals.” Nowhere in the mutation data is ALE duration described by 7 time intervals.

Response: Now added under the Methods section (line 431).

We would like to thank the reviewer for their detailed and thorough review!

Reviewer #2 (Remarks to the Author):

This paper describes a meta-study of E.coli evolution experiments, in which mutation data under different conditions of stress and adaptive evolution are harvested for a predictive framework based on machine learning methods. The novelty and strong point of this paper is the combination of data from different experiments into a unifying analytical framework, which drastically increases the information about repeatable mutation patterns and lays the ground for training predictive methods. At the same time, this strategy is what raised most of my questions on the paper, which are on the justification of pooling data for specific inference purposes and on the validity of some results. I think the points below should be carefully addressed in a revision.

1. Increase of substitution rate with time. This point, shown in fig. 3c, puzzled me most and I have difficulties in rationalising it. Most adaptive evolution experiments can be expected to take place under clonal interference, where the rate of substitutions is nearly constant and their fitness effects are a slowly decreasing function of time (this is the case, for example, in the Lenski experiment). To analyze the pattern observed here: Is the same pattern observed for synonymous and non synonymous changes? For the long term experiments, there should be data at intermediate time points; do they follow the same pattern? I am worried that the long-term experiments may target systematically other strains or selective stresses, which can confound the results.

Response: We agree with the reviewer; the linear pattern is what is expected to see and in fact this is the actual patterns when hyper-mutators and not taken into account. To clarify this point, we have updated Fig. 3c, by removing the hyper-mutators (which are included in the inset plot for completeness). In addition, we evaluated the synonymous and non-synonymous substitutions, as suggested, and we observe a similar linear pattern (please see supplementary Fig. 6a and 6b). The same linear pattern is observed for the number of synonymous and non-synonymous substitutions in the intermediate time plots, as well as for deletion and insertion mutations (Supplementary Fig 6c and d). We have also added this clarification in the manuscript (lines 156-158).

We also investigated if there is any systematic bias due to the experimental settings, so we conducted ANOVA analysis to elucidate this point. We split all the evolution runs into two groups according to one of the following factors: generation, strain, medium and stress. These results are shown in Supplementary Fig 7 and discussed in lines 174-176.

2. *Repeatability of mutational patterns.* The pattern observed in Fig. 4a-c can probably be digested in a more informative probabilistic model giving the likelihood to mutate a given gene in a given condition, such that the enhancement of repeatability over a null model of independent mutations (no repeatability) becomes clear. For example, a simple null model of independent mutations would give a frequency of $f = 1/k$ for a given mutation under k replicate experiments. The pattern of fig. 4a is a bit above that, $f \sim 1.5 / k$; how does that relate to the pattern of fig. 4b can 4c?

Response: To address this comment, we used linear regression to fit the pattern and also converted all plots in Fig. 4 to have the same independent variables, so the readers can compare. Fig. 4a now shows a relationship between the frequency (f) of a mutation within a condition and the reciprocal of the number of replicates ($1/k$) and the black line drawn corresponds to the linear regression line of $f=1.49/k + 0.018$ (and the reviewer is spot on $f \sim 1.5/k$). Similarly, the global overlap ratio (g) shown in Fig. 4b decreases as k increases and here we have $g= 1.5/k + 0.015$, which is very similar to the relationship between the averaged frequency and the reciprocal of the number of replicates. We include the null model (each mutation present in only one replicate, which leads to $f = 1/k$, where k is the number replicates) as a dotted line in Fig. 4a, with the size of the difference between the dashed and the black line representing the common mutations shared across replicates under any given condition. We have added this analysis in the manuscript (lines 193-196, 203-209).

3. *Inference of epistasis.* The analysis of co-occurrence and epistasis on p.8. appears to have an inconsistency. Epistasis means that the selection on one mutation depends on the genetic background it appears on, so only subsequent mutations in the same evolving population can be counted and the data must not be pooled across populations (cf. Methods, line 403-404). I realise that the data may not be informative enough under this more stringent requirement; the analysis of epistasis is not central to the paper and could also be left out without too much loss.

Response: We agree with the reviewer and to keep it clear, we removed the epistasis analysis and kept the co-occurrence analysis only (lines 196, 301-323).

4. *Predictive methods.* The discussion of this central point of the paper is too short and too much of a black box. It would be interesting to know the consistency of predictions from the different methods. A naive look at fig. 5b suggests that the other methods do not add much to a naive Bayes prediction, perhaps the main analysis should be limited to that method? What happens if more challenging separations into training and validation data are applied? Can we learn informative and biologically meaningful clusters of features, i.e., of strains, media, and stress conditions? Key features of the prediction methods should be described in Methods and Supporting Information to make the paper reasonably self-contained for a non-specialist readership.

Response: We have now added information about the optimization process in the manuscript (lines 398-401) and in the Supplementary material (Table 5, 6 and 7). The consistency of the predicted mutations by the three methods is summarized in the supplementary Fig. 9, where we have calculated and represented by a Venn diagram the overlap of the predictions for the different methods (line 245). We investigated how the performance changes for more challenging separations by performing a 10-fold cross validation, where we found a moderate reduction to AUPRC (0.28 vs 0.37) and almost no change to AUC (0.92 vs. 0.95), we include these results in lines 252-253. Regarding the performance of the various methods, it is true that the Naïve Bayes method is performing surprisingly well, but still the Ensemble method is significantly better (0.37 vs 0.18 in AUCPR). Regarding the information carried by each feature or groups of features, we have provided a ranked list of features in Supplementary file S8 and we have also revised the feature analysis in the manuscript to be more clear (lines 223-225).

4. *Minor points:* (a) Figs. 2cd would be more convincing on a log-log scale. (b) The discussion can be

improved in its coherence. For example, the second and third paragraph on p. 16 list a number of statements on evolution; I am not sure if all of them reflect the intention of the papers cited and how these statements are linked to the central results of the paper. A critical discussion of the state of predictions, including the current level of accuracy and ways for improvement, would be more appropriate.

Response: (a) Although we agree log-log is better in general, but we don't think it is appropriate for Fig. 2c as x-axis is indices. For Fig. 2d, we now include the log-log plot as an inset. (b) We have revised the discussion section to be more representative of the state of predictions and potential improvements (lines 300-323).

Thank you for your comments!

Reviewer #3 (Remarks to the Author):

This manuscript describes the significant effort undertaken by Wang et al. to survey, compile and curate E. Coli mutation events under different environmental settings. Except for a few minor errors, I found the paper well written with a logical flow and interesting findings. The major contribution of this paper is an "ensemble mutation predictor" that predicts which gene/intergenic regions are going to be mutated under a certain condition. I have a few questions and I think they might help me and the readers understand the method

better:

1. The authors mention that there are 83 features and they are shown as binary (T/F) individual features in Fig. 5A. (This pane is a bit confusing and might imply that there are 3 sets of features.) It seems that a couple of these features are categorical (e.g., Strain, Medium and Time) and should be encoded as 1-of-K. That is, only one Strain feature can be True and all other Strain features must be False. Stress seems to be different though, as more than one Stress feature can be True. I think feature encoding should be clearly defined and the number of features be updated (all Strain features count as one). Moreover, feature selection should be done differently on categorical features (all levels must be treated jointly).

Response: We have now changed Fig 5a to better reflect the grouping of the 83 features together. The reviewer is correct, we are using 1-K embedding for strain and medium, while in the case of strain features, it is still binary encoding with multiple non-zero entries allowed. We have added this clarification in the Method part (line 332).

2. If I understand correctly, there are 1,990 models for 1,990 gene/intergenic regions and all these models are trained independently, whereas mutations observed are presumably dependent. What would be the rationale for assuming independence and how this assumption can be verified/tested?

Response: Correct, we assume independence of mutations in this work, the existence of one mutation does not affect the probability of other mutations being present. This is clearly a simplification and future work should address this. We calculated the chi-square test on the null hypothesis model of independent mutations and report it in Supplementary Fig. 11. Another potential improvement to the model would be to add a time series component to the analysis, i.e. adding the order by which mutations happen, as we believe that some mutations are bound to happen earlier in the evolutionary trajectory, given an environment. We have added these points in the discussion section, lines 335.

3. Since there are 1,990 models and 1,990 sets of performance metrics, reporting metrics must include their statistical variation. For example, in Fig. 5B, the ROC curves are averages over all models. Which one is the best-performing model and which one is the worst-performing model? What is the variation in AUC and AUPRC (median, SD, etc)? (On a side note, the title of Fig. 5C is incorrect). I think boxplots of AUCs and AUPRCs could be informative. Moreover, a statistical test must be performed to show

superiority of one model over another. In Fig. 5C, the difference between ANS and all underlying models is huge. What is the explanation? Also, I do not think PRCs can be averaged.

Response: The distribution of AUC and AUPRC for each model and the ensemble predictor have been added in Figure 6 c and d. In addition, we have added the confusion matrices for all models in Supplementary file S9. The variation in the AUC and AUPRC of the ensemble predictor is AUC: 0.95 ± 0.06 , AUPRC: 0.37 ± 0.19 and we have included this in the manuscript (line 246). We have corrected the title in Fig. 5c. We agree a boxplot is informative to show the distribution and we used boxplot in Fig 6 c and d. We have calculated the statistical significance of the performance difference across models (t-test) and included this statistic in Table 4 of the Supplementary Materials. Although we did initially, we now do not average the PR curves, instead we flatten the prediction matrix for all genome sites, which yields similar results.

4. The frequency of mutation at each region is very low and the training problem is extremely imbalanced. How is this incorporated in the training process? I would recommend bootstrap aggregating (bagging) combined with oversampling of the rare class.

Response: The reviewer is right that imbalance is an issue in the dataset and we have used bootstrapping with oversampling when training the predictors to alleviate this issue. The rare class was oversampled to 20-50% of the whole dataset. This resulted in a slight increase of the AUPRC (Table 3 of Supplementary Materials). Other methods that can be used are the ADASYN (reference 117) and SMOTE (reference 118) techniques, which create new synthetic samples of the minority class (mutation class here), that reside close to the boundary of the majority and minority classes, so the predictor becomes more sensitive to minute differences of the samples that lay within the margin within the two classes. We now report the results of addressing class imbalance in line 231 to 233.

5. Why does the Naïve Bayes model perform so much better than the other two very flexible models? The ANN model seems very simple (only 6 nodes per layer). Are modern neural net frameworks such as TensorFlow or PyTorch are being used?

Response: The hyper-parameters for an ANN have been optimized by a 3-step procedure using Keras with Tensorflow as the backend. First, the activation function for the hidden layer and optimization method for training a neural network were optimized based on recommendation in literature (reference 110-112 in the manuscript) and our experimental results. Then the number of nodes in hidden layer and number of layers were optimized by the random search (reference 113 in the manuscript). Finally, other activation functions and optimization methods which were not chosen in step 1 were tested again (Results are in Tables 5-7 in SOM). We agree that the Naïve Bayes is performing surprising well, still less than the ANN but not significantly so, with the Ensemble classifier significantly outperforming all others. We expect that more flexible methods, such as ANN, will become increasingly more effective as the number of samples in the database increases and the minority class has sufficient data so that underlying non-linear relationships can be capture effectively.

6. The ROC curve in Fig. S5A is not accurate. In general, when there are few data points or there are ties in the predicted scores, one must bootstrap to report uncertainty. That is, sample n points with replacement, compute AUC and AUPRC, repeat 10 or 100 times and report mean AUC and AUPRC and their SDs.

Response: We have now used 10x bootstrapping and included the results (mean +/- STD) in the manuscript (line 272; AUC: 0.69 ± 0.08 , AUPRC: 0.17 ± 0.03).

We would like to thank the reviewer for his insightful comments, especially on the computation part of the work!

Reviewer #1 (Remarks to the Author):

All items from the previous review have been addressed in the recent manuscript revision. The authors describe the prediction model code used in this publication to be available on their website (line 80), which it currently is not.

All questions from the third reviewer were addressed by the authors except for the following: "In Fig. 5C, the difference between ANS and all underlying models is huge. What is the explanation?"

This question remains valid and no clear response was included by the authors.

Reviewer #2 (Remarks to the Author):

I have studied the revised version and the authors' reply to the referees' comments. Most of my comments have been addressed in a satisfactory way. In particular, the degree of repeatability of mutations (fig.4) now becomes clearer and the relative contributions of the predictor methods are transparent. I can recommend the manuscript for publication in this form.

We would like to thank the reviewer 1 for careful and thorough reading of our work. We have responded each point brought by the reviewer as follows (Each response follows the corresponding remark by the reviewers; remarks are in italics)

Reviewer #1 (Remarks to the Author):

1. The authors describe the prediction model code used in this publication to be available on their website (line 80), which it currently is not.

We have updated the data availability statement and we added the predictor module in our website: <http://www.mutationdb.com/CultureCondition>. The website allows a user to define a culture condition and generates a downloadable file which includes the probability of each of the 1990 genome sites hit by a mutation under the defined condition.

2. All questions from the third reviewer were addressed by the authors except for the following:

"In Fig. 5C, the difference between ANS and all underlying models is huge. What is the explanation?"

This question remains valid and no clear response was included by the authors.

The difference between the Ensemble method (AUPR = 0.36) and the best individual predictor, the Feed-forward ANN (AUPR = 0.32), is not considered large (less than 15%) and it is common that the Ensemble predictor to have an incremental improvement to predictive performance as it leverages information from all the individual predictors, which can be superior due to the diversity of the underlying models [1]. The Artificial Neural Network and Support Vector Machine are discriminative models and the Naive Bayes is a generative model. Besides it has been proved that stronger predictors can be built out of weak predictors [2].

References:

[1] Kuncheva, L.I., Whitaker, C.J.: Measures of diversity in classifier ensembles and their relationship with the ensemble accuracy. *Machine Learning* 51(2) (2003) 181–207.

[2] Schapire, R.E.: The strength of weak learnability. *Machine Learning* 5(2) (1990) 197–227